# D-DOPA Is a Potent, Orally Bioavailable, Allosteric Inhibitor of Glutamate Carboxypeptidase II

**DOI:** 10.3390/pharmaceutics14102018

**Published:** 2022-09-23

**Authors:** Sadakatali S. Gori, Ajit G. Thomas, Arindom Pal, Robyn Wiseman, Dana V. Ferraris, Run-duo Gao, Ying Wu, Jesse Alt, Takashi Tsukamoto, Barbara S. Slusher, Rana Rais

**Affiliations:** 1Department of Neurology, Johns Hopkins School of Medicine, Baltimore, MD 21205, USA; 2Johns Hopkins Drug Discovery, Johns Hopkins School of Medicine, Baltimore, MD 21205, USA; 3Department of Pharmacology and Molecular Sciences, Johns Hopkins School of Medicine, Baltimore, MD 21205, USA; 4Departments of Psychiatry and Behavioral Sciences, Johns Hopkins School of Medicine, Baltimore, MD 21205, USA; 5Department of Oncology, Johns Hopkins School of Medicine, Baltimore, MD 21205, USA; 6Department of Neuroscience, Johns Hopkins School of Medicine, Baltimore, MD 21205, USA; 7Department of Medicine, Johns Hopkins School of Medicine, Baltimore, MD 21205, USA

**Keywords:** catechol, glutamate carboxypeptidase II, CNS, D-DOPA, pharmacokinetics, brain penetration

## Abstract

Glutamate carboxypeptidase-II (GCPII) is a zinc-dependent metalloenzyme implicated in numerous neurological disorders. The pharmacophoric requirements of active-site GCPII inhibitors makes them highly charged, manifesting poor pharmacokinetic (PK) properties. Herein, we describe the discovery and characterization of catechol-based inhibitors including L-DOPA, D-DOPA, and caffeic acid, with sub-micromolar potencies. Of these, D-DOPA emerged as the most promising compound, with good metabolic stability, and excellent PK properties. Orally administered D-DOPA yielded high plasma exposures (AUC_plasma_ = 72.7 nmol·h/mL) and an absolute oral bioavailability of 47.7%. Unfortunately, D-DOPA brain exposures were low with AUC_brain_ = 2.42 nmol/g and AUC_brain__/plasma_ ratio of 0.03. Given reports of isomeric inversion of D-DOPA to L-DOPA via D-amino acid oxidase (DAAO), we evaluated D-DOPA PK in combination with the DAAO inhibitor sodium benzoate and observed a >200% enhancement in both plasma and brain exposures (AUC_plasma_ = 185 nmol·h/mL; AUC_brain_ = 5.48 nmol·h/g). Further, we demonstrated GCPII target engagement; orally administered D-DOPA with or without sodium benzoate caused significant inhibition of GCPII activity. Lastly, mode of inhibition studies revealed D-DOPA to be a noncompetitive, allosteric inhibitor of GCPII. To our knowledge, this is the first report of D-DOPA as a distinct scaffold for GCPII inhibition, laying the groundwork for future optimization to obtain clinically viable candidates.

## 1. Introduction

Glutamate carboxypeptidase II (GCPII), also known as prostate-specific membrane antigen (PSMA), is a type II transmembrane metallopeptidase encoded by the folate hydrolase (FOLH1) gene in humans [1]. Since its discovery in 1987 [2], its expression in different tissues such as the prostate, kidney [3], small intestine [4,5], and central and peripheral nervous system [6,7,8] has been reported. In the brain, GCPII catalyzes the hydrolysis of the neurotransmitter *N*-acetylaspartylglutamate (NAAG) to *N*-acetylaspartate (NAA) and glutamate [9], and multiple independent research groups have demonstrated the therapeutic benefit of inhibiting GCPII to treat neurological dysfunctions [6,7,8]. Similarly, perturbation in GCPII expression has been reported in cancer neovasculature [10], tumor angiogenesis [11], as well as in inflammatory bowel disease [12,13,14]. GCPII inhibition is currently being explored as a therapeutic modality for these diseases [6,15,16].

Structurally, nearly all GCPII active site inhibitors have similar pharmacophoric requirements; the scaffold consists of a zinc-binding group, a linker, and a carboxylic acid-containing moiety designed to interact with the S1′ glutamate recognition site of the enzyme [9]. The most extensively explored zinc-binding groups in the design of GCPII inhibitors include phosphonates/phosphinates, ureas, hydroxamates, and thiols [9]. Given this, GCPII inhibitors designed to date are highly charged, with poor PK and limited brain penetration [17,18,19,20].

One strategy to overcome these limitations is to design prodrugs to mask the hydrophilic sites and thus improve bioavailability [19,20,21]; for example, our group has reported a five-fold improvement in rodent plasma levels of 4-carboxy-α-[3-(hydroxyamino)-3-oxopropyl]-benzenepropanoic acid, a potent hydroxamate-based GCPII inhibitor, by masking its hydrophilic hydroxamate site using *para*-acetoxybenzyl-esters [20]. Similarly, oral administration of tetraODOL, and tris-POC prodrugs of the potent GCPII inhibitor 2-(phosphonomethyl)-pentanedioic acid (2-PMPA), yielded significantly improved plasma exposure compared with equimolar 2-PMPA in dogs and mice [18,21]. Intranasal (IN) administration of GCPII inhibitors has also been explored, specifically to improve their brain penetration index [22,23]. Nedelcovych et al. reported a consolidation of the two strategies by assessing the brain penetration of γ-(4-acetoxy benzyl) ester prodrug of 2-PMPA intranasally and showed that intranasal (IN) administration of the ester prodrug more than doubled 2-PMPA concentrations in the cerebrospinal fluid [22]. Lastly, dendrimer-based delivery systems have been employed to facilitate brain-targeted delivery of GCPII inhibitors to activated microglia [24,25]. Although these strategies have been somewhat promising in improving plasma and brain exposures, the search has continued for oral brain penetrable small-molecule inhibitors. 

Catecholic moieties have been reported to bind to zinc metalloproteases by various groups [26,27]; therefore, we evaluated known catechols as possible GCPII inhibitors. Systematic characterization led to identification of D-DOPA as the most promising catechol-based inhibitor. We herein report a detailed analysis of the oral bioavailability, brain penetration and GCPII target engagement of orally administered D-DOPA and demonstrate improvement when co-administered with the D-amino-oxidase inhibitor sodium benzoate. Moreover, we characterize the allosteric mode of GCPII inhibition of D-DOPA. To our knowledge, this is the first report of a catechol-based GCPII inhibitor; further evaluation and optimization may aid in the discovery of a candidate for clinical translation.

## 2. Materials and Methods

### 2.1. Reagents and Chemicals 

D-DOPA, L-DOPA, caffeic acid, L-DOPA-*ring-d*_3_, acetic anhydride, and compounds **3**, **5**, and **7**–**9** were purchased from Sigma-Aldrich (St. Louis, MO, USA) and used without further purification. Compounds **4** and **6** were synthesized using the previously reported methods [28,29]. LC-MS grade water, methanol, acetonitrile, and formic acid were obtained from Thermo Fisher Scientific (Waltham, MA, USA).

### 2.2. GCPII Activity Assay 

Inhibition potencies against GCPII (IC_50_ values) were determined using previously described methods with minor modifications [30]. Briefly, reactions were carried out in the presence of NAA-[^3^H]-G and human recombinant GCPII enzyme in Tris-HCl and CoCl_2_ at 37 °C for 20 min. Reactions were stopped with ice-cold sodium phosphate buffer containing 1 mM EDTA. Aliquots were then transferred to 96-well spin columns containing AG1X8 ion-exchange resin and centrifuged. NAA-[^3^H]-G was bound to the resin and [^3^H]-G eluted in the flow-through. Columns were washed with formate to ensure complete elution of [^3^H]-G. The flow-through and the washes were collected, and aliquots were transferred and dried to completion in a solid scintillator-coated 96-well plate. The radioactivity corresponding to [^3^H]-G was determined with a scintillation counter. Subsequently, IC_50_ curves were generated from CPM results.

### 2.3. Metabolic Stability Assays 

The metabolic stability of caffeic acid and L- and D-DOPA was evaluated in mouse plasma and brain homogenate using methods described previously by our group [31]. Briefly, brain homogenates were prepared by adding 9X of 0.1 M potassium phosphate buffer to weighed tissue and homogenization using a probe sonicator. Plasma and crude brain homogenate (1 mL each) were spiked with the analyte to give a final concentration of 10 µM and then incubated at 37 °C for 1 h. At predetermined time points (0, 30, and 60 min), an aliquot from each matrix was quenched with 3X volume of acetonitrile containing the internal standard (IS; losartan: 0.5 μM) and vortex-mixed for 30 s. The final mixtures were centrifuged at 10,000× *g* for 10 min at 4 °C and supernatants were used for LC-MS analyses.

The samples were analyzed using a Dionex Ultimate 3000 ultra-high-performance LC system coupled with a Q Exactive Focus orbitrap mass spectrometer (Thermo Fisher Scientific Inc., Waltham MA, USA) operated with a heated electrospray ionization (HESI) ion source. Each respective analyte was separated using the Agilent Eclipse Plus C18 column (2.1 × 100 mm i.d.; 1.8 µm) that was maintained at 35 °C while pumping a flow of 0.4 mL/min for 9 min using gradient elution of the mobile phases consisting of water and acetonitrile; both solvents contained 0.1% formic acid. The mass spectrometer was operated in switching ionization mode to collect both positive and negative molecular ions while being controlled by Xcalibur software 4.0.27.13 (Thermo Scientific). Analytes were quantified and the disappearance of the respective catechols was monitored in the full-scan mode (from *m*/*z* 50 to 1600) by comparing samples at the different time-points.

Metabolites were identified from these spectra by comparing samples quenched at 0 vs. 30 min. Accurate *m*/*z* values obtained from this analysis were used to propose metabolite structures. 

### 2.4. Pharmacokinetic Studies in Mice 

All of the animal studies were performed as per protocols approved by the Institutional Animal Care and Use Committee at Johns Hopkins University.

For in vivo PK studies, male CD-1 mice (Harlan Laboratories, Indianapolis, IN, USA) weighing 25–30 g were used. The mice were maintained on a 12 h light−dark cycle with ad libitum access to food and water. For initial screening, the mice were dosed perorally (PO) with a freshly prepared formulation of either caffeic acid (in 10% ethanol/10% tween/80% PBS *v*/*v*/*v*), L- DOPA or D-DOPA (in PBS) to administer a 50 mg/kg dose (at a dosing volume of 10 mL/kg) of the respective catechol. The mice were euthanized using CO_2_ at 30 and 60 min post drug administration; blood samples were collected in heparinized microtubes by cardiac puncture and spun at 2000× *g* for 15 min to collect plasma and then immediately frozen and stored at −80 °C. Brains were dissected and immediately flash frozen (−80 °C).

For full PK studies, D-DOPA was dissolved in PBS and administered either PO at a dose of 50 mg/kg or intravenously (IV; 10 mg/kg). For combination studies with sodium benzoate, a separate cohort was dosed with 400 mg/kg sodium benzoate IP, 5 min prior to D-DOPA (50 mg/kg PO). All of the formulations were freshly prepared prior to the dosing. The mice were sacrificed at specified time points (0.08, 0.25, 0.5, 1, 3, and 6 h) post drug administration. Brain and plasma collection were carried out as described above.

### 2.5. Bioanalysis in Plasma and Brain

Quantification of caffeic acid and L- or D-DOPA was performed using sensitive and selective methods reported in the literature with some modifications [32,33]. Caffeic acid was analyzed underivatized. For caffeic acid analysis, calibration standards were prepared by spiking standard solutions (in methanol containing 0.5 µM losartan) into naïve plasma or brain extract (prepared in methanol containing 0.5 µM losartan). Plasma and brain tissue samples were processed by matching dilutions of the calibration standards. Plasma samples were vortex mixed and brain samples were homogenized in a Geno grinder for 3 min at 1500 cycles per minute followed by centrifugation at 10,000× *g* for 10 min at 4 °C to collect the supernatant and 50 μL aliquots of the supernatant were transferred to 250 μL polypropylene autosampler vials sealed with teflon caps. Then, 2 μL of the sample was analyzed using LC/MS/MS system. 

L- and D-DOPA were analyzed using a previously described analytical method employing derivatization with an anhydride [32]. Briefly, for L-/D-DOPA analysis, calibration standards were prepared by spiking standard solutions of the analyte (in acetonitrile containing 0.1% formic acid) and internal standard (L-DOPA-*ring-d*_3_; 1 µM in acetonitrile containing 0.1% formic acid) in either plasma or brain extract (in acetonitrile containing 0.1% formic acid). Plasma and brain tissue samples were processed by matching the dilutions of calibration standards of the respective tissues. Plasma samples were vortex mixed and brain samples were homogenized in a Geno grinder for 3 min at 1500 cycles per minute followed by centrifugation at 10,000× *g* for 10 min at 4 °C to collect the supernatant. To the supernatants, sodium bicarbonate (0.2 M; pH = 8.3) was added followed by derivatizing solution (acetic anhydride in equal volume acetonitrile). The solutions were vortexed for 15 s and then allowed to sit at room temperature for 45 min. These mixtures were vortexed again for 10 s and centrifuged at 10,000× *g* for 10 min to collect the supernatant. Then, 2 μL of the sample was injected into the LC/MS/MS system for analysis.

For D-DOPA we also conducted intra-day and inter-day precision and accuracy tests for QC samples in plasma and brain (for inter-day, n = 3/day over 2 days), and tabulated statistical estimates. In addition, we evaluated autosampler stability at 4 °C by analyzing QC samples (n = 3/each QC level) immediately after sample preparation and after storage on the benchtop or in the autosampler for 18 h. To assess carryover between injections we injected a double blank sample immediately after the highest standard in the calibration curve (upper limit of quantitation-ULOQ; 100 nmol/mL in plasma). 

Chromatographic analysis was performed using a Thermo Scientific Vanquish UPLC system consisting of an analytical pump and an autosampler coupled with a TSQ Altis mass spectrometer. For all analytes, the mobile phase used for the chromatographic separation consisted of 0.1% formic acid in acetonitrile and 0.1% formic acid in water. The mobile phase was delivered as a gradient at a flow rate of 0.400 mL/min. Separation of the analyte was achieved using an Agilent EclipsePlus C18 RRHD, 1.8 µm (2.1 mm × 10 mm) column. The analyte was monitored using a ThermoScientific TSQ Altis triple-quadrupole mass-spectrometric detector equipped with an electrospray interface, and operated in negative ion mode. The instrument was controlled by Thermo Scientific Xcalibur (version 4.2.47) software. The spectrometer was programmed in selected reaction monitoring (SRM) mode to monitor the transitions for caffeic acid *m*/*z* 179.0 → 107.1, 135.1 and losartan *m*/*z* 421.1 → 127.1, 179.1, for L- and D-DOPA *m*/*z* 322.0 → 238.1, 260.0, 262.1, for L-DOPA-*ring-d*_3_ *m*/*z* 325.0 → 241.1, 265.1, 283.0.

For PK analyses, plasma levels (nmol/mL) and brain tissue concentrations (nmol/g) were determined and plotted against time, and then non-compartmental analysis modules in Phoenix WinNonlin version 7.0 (Certara USA, Inc., Princeton, NJ, USA) were used to quantify exposures (AUC_0–t_), half-life (t_1/2_), volume of distribution (V_d_), and clearance (Cl).

### 2.6. D-DOPA Target Engagement Studies 

GCPII activity measurements were carried out based on a modification of a previously published protocol [34]. Briefly, brain samples were homogenized in ice-cold Tris buffer containing protease inhibitors. Resulting homogenates were spun down and the supernatants collected for both GCPII activity measurements and protein analysis. GCPII reaction was initiated upon the addition of homogenate, cobalt chloride, and ^3^H-NAAG (0.04 µM, 20 mCi/µmol). Reactions were carried out in 50 µL reaction volumes for 2 h at 37 °C. At the end of the incubation period, reactions were terminated with ice-cold sodium phosphate buffer and [^3^H]-glutamate measured as described above. Finally, total protein measurements were carried out using BioRad’s detergent compatible protein assay kit and data were presented as fmol/mg/h.

### 2.7. D-DOPA Mode of Inhibition Studies

To determine the mode of inhibition, enzymatic activity assays were carried out as described in the GCPII assay section above [15,30,34], except that concentrations of NAA-[^3^H]-G were extended up to 4 µM and in the presence and absence of D-DOPA (0, 100, 200, 300, 400 nM). K_m_, V_max_ calculations were determined using Michelis-Menten kinetic analysis using GraphPad Prism (version 9.3.0, Dotmatics, San Diego CA, USA).

## 3. Results and Discussion

### 3.1. IC_50_ of Catechol-Based Scaffolds

The inhibitory potencies of the catechols were determined using a modified radioactivity-based assay involving human recombinant GCPII enzyme as previously reported [30]. Based on literature reports of catecholic moieties binding to zinc metalloproteases [26,27], we tested the potency of L-DOPA (**1**) against GCPII activity and discovered that it is a submicromolar inhibitor. Although its potency is weaker than those of known GCPII inhibitors containing multiple carboxylate groups, L-DOPA represents one of the most potent monocarboxylate-based GCPII inhibitors. This prompted us to evaluate analogs of L-DOPA (Table 1) to determine the essential structural features responsible for its GCPII inhibitory activity. Both removal of the meta-hydroxy group and methylation of the two hydroxy groups resulted in substantial loss of potency as seen in compounds **2** and **3**. Replacement of the catechol moiety with a 2-pyridone (compound **4**) also led to complete loss of potency. These findings suggest that the catechol moiety of L-DOPA is essential for the GCPII inhibitory activity. Subsequently, we tested L-DOPA analogs retaining the catechol moiety (compounds **5**–**10**). Alpha-methylation (compound **5**), *N*-acetylation (compound **6**), and decarboxylation (compound **7**) resulted in loss of potency, whereas the removal of the alpha-amino group (compound **8**) led to only a 6.7-fold decrease in inhibitory potency. Interestingly, conversion of the alpha–beta single bond of compound **6** to a double bond (compound **9**, caffeic acid) generated another potent GCPII inhibitor with an IC_50_ value of 300 nM. Finally, D-DOPA, an enantiomer of L-DOPA, was found to be the most potent catechol with an IC_50_ value of 200 nM. Taken together, these structure–activity relationship (SAR) studies highlight the critical role played by the catechol and carboxylate groups of L-DOPA in GPCII inhibition as well as the insignificant contribution of the alpha amino group. We chose to further assess the three submicromolar inhibitors (L-DOPA, caffeic acid, and D-DOPA) for their potential to serve as in vivo pharmacological probes for GCPII inhibition.

### 3.2. In Vitro Metabolic Stability of Caffeic Acid, L-DOPA, and D-DOPA

Our primary objective was to identify catechols that would resist systemic and CNS metabolism to ensure biodistribution to the central nervous system (CNS) where the compounds could then inhibit GCPII activity. Caffeic acid, L-DOPA, and D-DOPA were first screened in vitro, both in plasma and in brain homogenate (Figure 1) to assess their metabolic stability. Caffeic acid was found to be stable (98% remaining) in brain homogenate when incubated at 37 °C for up to 1 h post spiking; whereas in plasma, it was found to be partially metabolized (56% remaining at 60 min; Figure 1a). Although some instability was observed in plasma, no metabolites were identifiable in metabolite identification (MET-ID), perhaps due to formation of smaller, polar metabolites not amenable to reverse-phase LCMS analyses. L-DOPA was stable in plasma (64% remaining at 60 min) and unstable in brain (0% remaining by 30 min; Figure 1b). In contrast, D-DOPA was stable in plasma (100% remaining at 60 min) and brain (82% remaining at 60 min; Figure 1c). The metabolism of L-DOPA to dopamine by aromatic L-amino acid decarboxylase (AAAD) is well reported [35], and thus we expected L-DOPA to be rapidly metabolized in brain homogenate. This was further confirmed by MET-ID which showed significant dopamine levels in brain homogenate at 30 min (Appendix A). The moderate stability of L-DOPA in plasma could be attributed to the activity of AAAD expressed in endothelial cells of blood vessels. However, no dopamine was observed in L-DOPA-spiked plasma samples, perhaps due to rapid conversion to downstream metabolites [36,37]. The D-enantiomer, D-DOPA, is not recognized or metabolized by AAAD and thus was observed to be completely stable in both plasma and brain and qualitative MET-ID studies did not reveal dopamine increases in D-DOPA-spiked brain homogenate.

### 3.3. Bioanalytical Methods for Caffeic Acid, L-DOPA, and D-DOPA

To conduct in vivo assessment of selected catechols we searched literature reports for facile and sensitive LC-MS methods. Caffeic acid resolved well on reverse-phase column and was detectable on MS with low nM sensitivity; however, reported methods for DOPA analyses either required complex sample preparation, such as solid phase extraction [38], or used harsh acidic conditions (e.g., 0.4 M perchloric acid) [38,39]. To circumvent this, we initially explored hydrophilic interaction liquid chromatography (HILIC) methods; however, these methods required longer run/equilibration times (>10 min) and afforded poor sensitivity (lower limit of quantitation; LLOQ > 100 nM). Furthermore, instability of DOPA at neutral pH presented additional challenges in sample preparation. DOPA is prone to oxidation and degrades over 4–8 h at room temperature forming melanin oligomer [40]. It was thus important to stabilize the analytes during sample preparation to prevent oxidation and also render them amenable for UPLC-MS analyses. We thus employed a derivatization method previously reported and validated by van Faassen et al., with minor modifications [32]. In addition, although many methods have been reported for DOPA analysis in plasma, few have evaluated brain levels. Given these challenges, we used acetic anhydride in the presence of sodium bicarbonate (pH = 8.3; 0.2 M) to acetylate the polar catecholic and amino groups thus improving the lipophilicity of DOPA for better retention on reverse-phase column (Figure 1) while also stabilizing the molecule for accurate analysis. For DOPA analyses, we used L-DOPA-*ring-d_3_* as an internal standard. DOPA and L-DOPA-*ring-d*_3_ were derivatized at three sites, including both phenolic sites as well as the primary amine of the amino acid [38]. The structures of acetylated L-DOPA, D-DOPA, L-DOPA-*ring-d*_3_, as well as underivatized caffeic acid, losartan, and their mass spectra and extracted chromatograms at the lower limit of quantitation are presented in Figure 2. The LLOQ for DOPA in plasma and brain was observed to be 0.03 nmol/mL and 0.10 nmol/g respectively; similarly, the LLOQ for caffeic acid in plasma and brain was observed to be 0.01 nmol/mL and 0.01 nmol/g respectively. These were similar or better than literature reports of LLOQ of L-DOPA in rat plasma, which varies from 0.02 to 0.15 nmol/mL [41,42], whereas for caffeic acid it is reported to be 0.03 nmol/mL [33]. A correlation coefficient of >0.99 was obtained in all analytical runs and internal standard variation of <15% was observed for all analyses. The mean-predicted accuracy for calibration standards ranged from 85 to 111% for caffeic acid, and 97 to 111% for L-DOPA. For quality control samples, the mean-predicted accuracy ranged from 86 to 98% for caffeic acid, and 86 to 113% for L-DOPA.

We next assessed the linearity, precision, accuracy of the LC-MS method, and benchtop and autosampler stability of derivatized D-DOPA, as some modifications were made to the method reported and validated by van Faassen et al. [32]. The results for these are presented in Appendix A. Acetylated D-DOPA was found to be stable on the benchtop and in the autosampler for up to 18 h with a <5% deviation from nominal concentrations in plasma. A blank injection following the highest standard of 100 nmol/mL in plasma revealed a minimal acceptable carry-over area of >1% for both the analyte and internal standard. The mean-predicted accuracy for calibration standards ranged from 93.7–112% and 93.6–115% in plasma and brain, respectively (Appendix A). Inter-day accuracy and precision (%RSD) of QCs (n = 3/day; 2 days) ranged from 95.9–107% and 3.15–5.40% in plasma, and 95.0–101% and 1.05–2.52% in brain, respectively (Appendix A). Thus, the LC-MS method used for D-DOPA analyses was found to be precise and accurate for the presented bioanalyses.

### 3.4. Initial Pharmacokinetic Studies of Caffeic Acid, L-DOPA, and D-DOPA in Mice

We evaluated caffeic acid and L- and D-DOPA in vivo in a two time-point PK study with the primary objective of selecting the compound that showed the best in vivo levels in plasma and brain for further analyses. Among the three catechols, D-DOPA showed ~22-fold higher plasma levels (84.3 nmol/mL; Figure 3c) compared with L-DOPA (3.74 nmol/mL; Figure 3b) and ~13-fold higher levels versus caffeic acid (6.47 nmol/mL; Figure 3a) at 30 min post dose. By one hour, the levels of both D- and L-DOPA had decreased, but D-DOPA maintained the highest concentration at 21.5 nmol/mL. In brain, L-DOPA and caffeic acid were below the limit of quantification (0.10 nmol/g for L-DOPA and 0.01 nmol/g for caffeic acid) at both time points; in contrast, D-DOPA showed quantifiable levels (~1 nmol/g) at both time points. These results clearly demonstrated the superiority of D-DOPA versus both L-DOPA and caffeic acid in terms of systemic and brain exposures following oral administration. 

Although caffeic acid was the most stable of the three analogs in metabolic studies in vitro, it showed low plasma levels (6–7 nmol/mL) following oral administration, which was consistent with its widely reported poor oral bioavailability and low intestinal absorption [43]. Furthermore, its hydrophilicity due to the presence of a negative charge limited its brain penetration. L-DOPA has also been widely studied previously and its clearance is expected to occur primarily via metabolism to dopamine during first pass metabolism [44,45]. To circumvent this, decarboxylase inhibitors such as carbidopa are routinely employed in combination with L-DOPA [46] and abundant evidence exists supporting higher peripheral and brain L-DOPA exposure when L-DOPA/carbidopa are co-administered [47]. For example, Diederich et al. demonstrated that co-administration of carbidopa significantly enhanced L-DOPA levels in rat brain [48] upon systemic administration (250 mg/kg L-DOPA IP with 40 mg/kg carbidopa) giving a brain/plasma ratio of 0.3–0.7 at 30 min post-dose. These preclinical data, as well as the approved use of this combination as a standard-of-care for Parkinson’s disease (PD) support further exploration of L-DOPA + carbidopa combination for GCPII inhibition; however, we advanced D-DOPA for further evaluation versus L-DOPA for the following reasons. First, chronic use of L-DOPA/carbidopa therapy is associated with neurotoxicity in preclinical models [49,50]; mechanistic studies show that L-DOPA neurotoxicity is related to dopamine generation and is not phenocopied with D-DOPA [51]. In PD patients, its long-term use is shown to cause disabling motor effects such as levodopa-induced dyskinesias (LIDs) and motor fluctuations in at least two thirds of patients [52,53]. Such side effects are not expected to occur with D-DOPA. Second, L-DOPA therapy would both increase dopamine and inhibit GCPII, whereas D-DOPA would not have a significant effect on dopamine and thus may inhibit GCPII more selectively. 

The off-target toxicity of L-DOPA, three-fold higher potency of D-DOPA versus L-DOPA, and superior in vitro stability of D-DOPA over other catechols justified the advancement of D-DOPA for further assessment as a potential GCPII inhibitor. 

### 3.5. Pharmacokinetic and Target Engagement Studies of D-DOPA + Sodium Benzoate in Mice

PK profiles of D-DOPA in mouse plasma and brain are presented in Figure 4a–c, and detailed PK parameters in Figure 4e. After a single IV dose of 10 mg/kg, D-DOPA achieved a maximum concentration (C_max_) of 61.1 nmol/mL in plasma at 5 min (T_max_) (Figure 4a). The half-life (t_1/2_), volume of distribution (V_d_), and clearance (Cl) of D-DOPA in plasma were calculated to be 0.35 h, 0.834 L/kg, and 28 mL/min/kg, respectively. The overall exposures in plasma (AUC_IV-plasma_) were calculated to be 30.5 nmol.h/mL (Figure 4e). After a single PO dose (50 mg/kg), D-DOPA achieved a C_max_ of 99.0 nmol/mL in plasma at 15 min and 1.74 nmol/mL in brain at 30 min (Figure 4b). AUC_PO-plasma_ and AUC_PO-Brain_ were calculated to be 72.7 nmol.h/mL and 2.42 nmol.h/g, respectively. D-DOPA also exhibited excellent oral bioavailability at 47.7% (AUC_PO/IV_). The brain penetration index was low with AUC_Brain/Plasma_ = 0.033. However, even with this low index, the concentrations of D-DOPA exceeded its IC_50_ up to 1 h. Several groups have reported that D-DOPA is unidirectionally converted to its L-isomer in vivo by oxidative deamination by the enzyme D-amino acid oxidase (DAAO) [54,55,56]; the resulting alpha-keto acid is then transaminated to L-DOPA by dopa transaminase. Thus, we strategized to evaluate the PK of D-DOPA when co-administered with a DAAO inhibitor as a mechanism to enhance its plasma and brain exposures.

Mice were dosed IP with 400 mg/kg of the DAAO inhibitor sodium benzoate, followed by PO administration of D-DOPA (50 mg/kg). D-DOPA achieved a 1.53-fold higher C_max_ (151 nmol/mL) in plasma at 15 min and a 1.84-fold higher C_max_ in brain (3.20 nmol/g at 30 min) compared with D-DOPA monotherapy (Figure 4b,c). Furthermore, plasma exposures (AUC_PO-plasma_) improved by 2.54-fold (185 nmol.h/mL; * *p* < 0.05) and AUC_PO-brain_ by 2.26-fold (5.48 nmol.h/g; * *p* < 0.05). Similar modulation in the PK of D-serine, a substrate of DAAO, has been reported by our group and others where D-serine combination with a DAAO inhibitor significantly decreased its clearance and increased its exposures [57,58]. The sodium benzoate combination with D-DOPA also enhanced the oral bioavailability of D-DOPA (47.7% as monotherapy versus >100% in combination) and was superior to known active site inhibitors with <5% bioavailability [59,60], with the exception of thiol-based GCPII inhibitor, that exhibit ~30–40% oral bioavailability in preclinical species [61]. 

The brain levels of D-DOPA, although low, remained above its IC_50_ value of 0.2 µM for over 3 h (Figure 4c). Lastly, we qualitatively assessed GCPII target engagement in the brain from D-DOPA monotherapy (C_max_ = 1.74 nmol/g) and combination therapy (C_max_ = 3.2 nmol/g), by evaluating ex vivo effects on GCPII enzymatic activity [62]. Both treatment groups showed significant inhibition of brain GCPII activity with the co-administration group having a superior effect (*** *p* < 0.001—untreated vs. combination therapy, ** *p* < 0.01—untreated vs. monotherapy; * *p <* 0.05—monotherapy vs. combination therapy corroborating the PK results (Figure 4d). We observed ~25% inhibition of GCPII although the concentration of D-DOPA was 10-fold higher than its IC_50_ value. This is likely due to the dilution of tissue during the ex vivo homogenization procedures which permitted the dissociation of D-DOPA from the GCPII enzyme. In addition, it is important to note that we measured total D-DOPA levels in the reported PK studies and thus target engagement results may be confounded due to binding of D-DOPA to other endogenous proteins. This off-target binding is expected to limit the availability of free D-DOPA concentrations at the target site, thus moderating the inhibition of GCPII. 

### 3.6. Characterization of D-DOPA’s Mode of GCPII Inhibition 

Most GCPII inhibitors designed to date contain pharmacophoric features for active site binding; viz., phosphonic acid-based [63,64], thiol-based [65], urea-based [66], and hydroxamic acid-based [67]. As discussed earlier, our initial premise for exploring the inhibition potencies of catechols against GCPII activity was their propensity for zinc binding in other similar metalloproteases [27]. Considering the structural differences between current active site inhibitors of GCPII versus catechols, we thought it important to evaluate the mode of inhibition.

To evaluate this, NAAG saturation experiments were performed in presence of different concentrations of D-DOPA (Figure 5a). When the rate of reaction was plotted against NAAG concentrations at increasing inhibitor concentrations, there was a decrease in maximal rate (V_max_) whereas the Michaelis constant (K_m_) was unchanged (Figure 5a). V_max_ and K_m_ for each dataset at a given inhibitor concentration were obtained from non-linear regression fits to Michaelis–Menten kinetics (Figure 5d). A double reciprocal plot (Lineweaver–Burk plot) of the data yielded lines with varying slopes that intersected in the second quadrant (Figure 5b), indicative of non-competitive inhibition. A secondary plot of K_m_ apparent/V_max_ versus [D-DOPA] gave a K_i_ value of ~400 nM (Figure 5c). Although the covalent binding of catechols, including L-DOPA and its analogs, to proteins via sulphahydryl interactions have been reported, this is the first description of a sub-micromolar, non-competitive inhibitor of GCPII. Thus, the mode of inhibition studies disproved our initial hypothesis and revealed D-DOPA to be an allosteric inhibitor of GCPII. Further, Parellada et al. tested various catechols and reported that the phenolic hydroxyl groups do not coordinate with the catalytic zinc of the active site in zinc metallopeptidases [68]. This report further supports the allosteric, non-competitive mode of inhibition demonstrated by our studies.

## 4. Conclusions

Small-molecule GCPII inhibitors containing glutamate-mimetics have been used as a treatment for neurological disorders in preclinical models [15,64,66,69,70,71,72,73,74,75,76,77]; however, their therapeutic potential in the clinic has been hampered, in part due to poor oral bioavailability and negligible brain penetration. To overcome these limitations, we assessed known catechols and identified three with submicromolar potencies for GCPII inhibition. This is the first report of the characterization of catechols as GCPII inhibitors. D-DOPA emerged as the most promising catechol-based inhibitor, demonstrating a non-competitive mode of inhibition and an excellent PK profile which was enhanced by co-administration with the DAAO inhibitor sodium benzoate, resulting in robust target engagement in the brain. To our knowledge this is the first systematic assessment of a submicromolar, catechol-based, allosteric inhibitor of GCPII. Future studies will focus on optimization of this scaffold for clinical translation of this new class of GCPII inhibitors.

## Data Availability

All data are contained within the article.

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
