# Peer review of "D-DOPA Is a Potent, Orally Bioavailable, Allosteric Inhibitor of Glutamate Carboxypeptidase II"

_pharmaceutics, 2022, doi:10.3390/pharmaceutics14102018_

Round 1

Reviewer 1 Report

The authors described in this manuscript the identification of L-, D-dopa, and caffeic acid as novel allosteric inhibitors for glutamate carboxypeptidase II. Potency, oral and brain availability makes D-dopa an interesting starting point for further optimization. Although the findings are very interesting for readers working in this field, there are a few major issues associated with the manuscript:

- The reason why D-dopa was preferred for further studies was not quite obvious. Though brain exposure of L-dopa was low after oral administration of the drug alone, there are plenty of data in the literature showing substantial brain exposure of L-dopa after oral treatment of L-dopa together with carbidopa. For example, Diederich et al. showed that L-dopa reached an exposure of about 4 µM in brain and a brain/plasma ratio of about 0.3 after an intraperitoneal injection of 250 mg/kg L-dopa and 40 mg/kg carbidopa in rat (Pharmacology 1997;55:109–116). Since L-dopa/carbidopa is an approved drug, it would be much easier to repurpose it than develop a new drug like D-dopa, A direct comparison between the combination of L-dopa/carbidopa and the combination of D-dopa/sodium benzoate would be more appropriate for judging the suitability of both drugs for further studies.

- The method used in this manuscript for the determination of in vivo target engagement doesn't seem to provide precise results (by the way, the reference 38 didn't contain the method as mentioned in this manuscript). Both L- and D-dopa seem to be reversible inhibitors, they will dissociate from the target during the preparation of the brain homogenate for the enzyme activity measurement. The inhibitory activities measured with the method described here didn't reflect the quantitative target engagement in vivo, but rather qualitative. This is obvious if one correlates the brain exposure at the time when brain samples were taken with the IC50 value and the degree of inhibition in the enzyme activity assay. The brain exposure of D-Dopa was 10-fold higher than its IC50 which should result rather in 90% inhibition, according to figure 4d, there was only a 20-30% inhibition.

Author Response

Please see the point-by-point response to rev.1

Reviewer 2 Report

As conventional inhibitors suffer from poor bioavailability and low brain exposure, this study investigated catechol-based compounds for the inhibition of glutamate Carboxypeptidase II. A lead compound, D-DOPA, was identified as having high potency, excellent in vitro stability, and a promising oral PK profile in plasma, yet suffered from poor brain penetration (3%).  To increase exposure to the brain, the group tested a combination therapy of D-DOPA and an inhibitor of a known metabolizing enzyme in vivo. This enhanced bioavailability in both plasma and brain tissue. The group concluding by elucidating D-Dopa’s mode of inhibition.

Overall, I thought the study was well written and contained quality science. I only have minor comments and would recommend this study for publication.

·      Introduction: last paragraph reads like a Conclusion…

·      As derivatization was ‘required’ for LC-MS analysis, an experiment to validate this approach would have increased my confidence in their quantification, albeit I don’t know if this is a standard procedure or not. I am also not convinced that acetylating the hydroxy/ amine groups was necessary in the first place. If the issue was compound retention on a C18 column, they could have used a column that didn’t rely on lipophilicity, and instead on aromatic interactions, since the compounds contained a phenolic Moiety.

·      Section 3.2: when referencing stability of D/L-DOPA and caffeic acid, instead of saying “completely stable”, please provide percent changes from baseline so the reader can better judge just how different the concentrations in plasma or brain are over 1 hr.  for example, D-DOPA slightly decreases in brain, but all you say is “mostly stable”.  That means nothing to the reader.  Was it -5%, -10%, or -20% (a more significant change)?

·      Remove “Time-dependent” from Figure 4 title.  All PK is dependent on time, but “time-dependent PK” implies something much different than you’re describing.  Time-dependent PK refers to changes in a drug’s PK profile over a long period of time (days-weeks) due to irreversible inhibition, upregulation, etc. 

·      An MTD experiment for their lead compound would have been informative but perhaps it escapes the scope of this publication.

·      Figures could use more labels, for ease of interpretation.

Round 2

Reviewer 1 Report

The authors didn't response to question 1 properly. In the response text, the authors correctly cited the brain exposure of L-dopa in rats from the publication by Diederich et al. (368 - 850 nmol/g tissue). However, the values used for figure a illustrating results from Diederich et al. were 3 oders of magnitude lower (0.3 - 1 nmol/g tissue). This led to the wrong calculation of the improvement factors for D-dopa in mice. The authors should also have looked up the exposure of L-dopa after oral treatment of l-dopa and carbidopa in human. Up to 10 nmol/ml maximal plasma exposure of l-dopa have been reported. Since L-dopa is reported to be crossing blood-brain barrier easily (in Diederich et al., plasma exposure of L-dopa is only 3-fold higher than brain exposure in rat), brain exposure of L-dopa in human after oral treatment with l-dopa and carbidopa easily reaches or exceeds the brain exposure of d-dopa in mice reported in this manuscript.

Round 3

Reviewer 1 Report

Current revision answered all questions.